# Friction and Wear Characteristics of Bacterial Cellulose Modified by Microcellular Foaming Process

Jin Hong [1] , Jae-Ho Han [1], Doyeon Kim [2], Gukhyeon Yun [1], Kwan Hoon Kim [1] and Sung Woon Cha [1,*]

[1] Department of Mechanical Engineering, Yonsei University, 50, Yonsei-ro, Seodaemoon-gu, Seoul 03722, Republic of Korea; jin.hong@yonsei.ac.kr (J.H.); jhhan9090@yonsei.ac.kr (J.-H.H.); gukhyeon.yun@yonsei.ac.kr (G.Y.); kimkevin99@yonsei.ac.kr (K.H.K.)

[2] Department of Food and Nutrition, Yonsei University, 50, Yonsei-ro, Seodaemoon-gu, Seoul 03722, Republic of Korea; do4312@yonsei.ac.kr

* Correspondence: swcha@yonsei.ac.kr; Tel.: +82-2-2123-4811

**Abstract:** Bacterial cellulose (BC) is a biodegradable, non-toxic, natural substance that can be obtained by culturing bacteria. It can be approached in various ways from physical, chemical, and biological points. BC nanoparticles have been applied as lubricating additives to improve the load capacity, anti-wear, and friction. The microcellular foaming process was created using a technology based on the saturation of the polymer by supercritical $CO_2$ and rapid decompression. An increase in saturation pressure leads to an increase in the molecular potential energy of $CO_2$, which can be more easily compressed into the cellulose matrix. Moreover, the high crystallinity and water content combination contribute to thermal stability. Specimen membranes produced by *Komagataeibacter xylinus* prepared with a thickness of 2 mm were saturated in supercritical condition, 10 MPa of $CO_2$ for 4 h, and foamed at a temperature of 120 °C in a hot press. After the foaming process, we used dry ice to cool the BC. Before foaming, the friction coefficient continuously increased with the increase in cycles, and after foaming, a stable friction coefficient of 0.3 or less was secured despite the increase in the cycle. The microcellular foaming process significantly reduced and made BC's coefficient of friction stable.

**Keywords:** microcellular foaming process; batch process; bacterial cellulose; coefficient of friction



## 1. Introduction

Biomedical polymer refers to a biocompatible polymer used to treat a disease, and the range of its industrial market is extensive and diverse. Based on the development of foams to date, theoretical predictions and the formation of cells inside the material on the physical properties of foam materials have shown various positive effects. The microcellular foam was also expected to exhibit excellent physical properties. It stands out in various fields, such as thermal insulation, low dielectric constant, improved mechanical performance, and improved optical transparency [1–5].

Biomedical polymers can be divided into biocompatible polymeric materials and cell- and tissue-compatible polymeric materials, and collectively refer to polymeric materials used for diagnosing or treating diseases [6]. Also, in common sense, it is a biocompatible polymer used to replace a body damaged by disease or accident, and various materials such as metals, ceramics, and composites are currently used. In particular, polymer materials that can be easily modified chemically or physically and can be used in various forms, such as tubes, films, and fibers, are most often used [7].

Recently, due to the rapid increase in the elderly population, degenerative diseases are making many patients sick worldwide, and these diseases have reduced the quality of human life. In particular, the number of patients undergoing surgery due to degenerative arthritis is increasing by 10% yearly. The knee area is the most representative of artificial

joint replacement among them. Many seniors are looking for a new life by inserting an artificial joint made of materials such as ceramic and metal into the body.

However, there is pain due to the shape mismatch between the knee and the artificial joint and discomfort due to the stiffness of the inserted joint. In addition, a bigger problem is the lifespan of artificial joints, which are limited to 10 years, and the problem of necrosis of surrounding tissues due to debris generated in the body due to long-term use of the artificial joint.

Various properties and basic conditions for using polymer materials for medical purposes are biocompatibility, temporary sterilization, appropriate mechanical properties, physical properties, and molding processability. Going further in detail, the most important feature is that it should be non-toxic, harmless, and safe even after the period of use or application to the human body [8,9]. Recently, many researchers have actively conducted research on the properties and characteristics of bacterial cellulose, such as modifying the surface of polymers or changing mechanical properties and properties to have an affinity with tissue cells in the human body. Mass production and use methods are also receiving significant interest [10,11].

In particular, in the case of medical devices under load, such as artificial joints, the physical and chemical characteristics of raw materials and parts are directly related to biological safety about the generation of particulates or the decomposition of substances due to wear, friction, fatigue, load-related, etc. In the case of bacterial cellulose, biocompatibility, biodegradability, and blood compatibility have been confirmed through previous studies. If the physical properties of the same material are improved without chemical or physical addition through the ultra-fine foaming method, the range of application will be much wider [6,12,13].

Bacterial cellulose is a film on a pellicle formed through the metabolism of bacteria during the culture of Acetobacter Xylinum and was discovered by Brown in 1886. The biosynthesis process occurs at the interface between the air requiring oxygen and the liquid medium in which the bacteria are dispersed. As a result, the three-dimensional structure of bacterial cellulose becomes more robust and thicker over time. Bacterial cellulose is a high molecular substance with the same chemical structure as plant cellulose and forms a thin and irregular network called microfibrils.

Bacterial cellulose has excellent physical properties such as high absorption, elasticity, moisture retention, crystallinity, and oxygen transfer efficiency, which were difficult to find in vegetable cellulose [14,15]. However, it is difficult to commercialize due to high production costs and difficulty in mass production. However, its applicability to biomedical and electronic materials is highly evaluated due to its high biocompatibility, flexibility, moisture content, and environmental friendliness. Moreover, currently, it is expanding into various product groups such as medical mats using high binding capacity, speaker diaphragms using high elastic modulus, thickening agents, swelling agents, and mask packs using high absorbency. It is being used as a high-value-added product as a biopolymer.

*Acetobacter Xylinum* (or *Gluconacetobacter xylinus*) was first described in 1886 by Adrian John Brown, who identified the bacterium while studying fermentation. Later, in 2013 by Yamada et al., a new genus *Komagataeibacter* was established and named *Komagataeibacter xylinum* (American Type Culture Collection, Virginia, USA, ATCC 23767). This is a species of bacteria best known for its ability to produce cellulose, specifically bacterial cellulose. Also, the strain is a member of acetic bacteria, a group of gram-negative aerobic bacteria that produce acetic acid during fermentation. Bacterial cellulose is involved in biofilm formation. It is chemically identical to plant cellulose but has a different physical structure and properties. The mechanical properties of polymers change depending on the degree of movement of polymer chains, but it is very difficult to change cellulose, a polysaccharide, without chemical treatment. However, $CO_2$, a physical blowing agent, can only make cells through solid-state foaming and foam extrusion. Therefore, in this study, we will check whether it is possible to produce a sheet that can be used as support applicable to the

human body by using a batch method that can preserve the cellulose sheet, as it is among various foaming methods.

## 2. Materials and Methods

The development of polymeric foams began in the 1930s, and the first patented foam was a macroporous polystyrene foam with a cell size of 100 μm or more in 1931. However, foams with finer cells are expected to provide better overall mechanical properties, so most foam studies have focused on increasing cell density and reducing cell size using various methods [16].

The foaming principle of the microcellular foaming process (MCP) is as follows regardless of the material or foaming method (Figure 1).

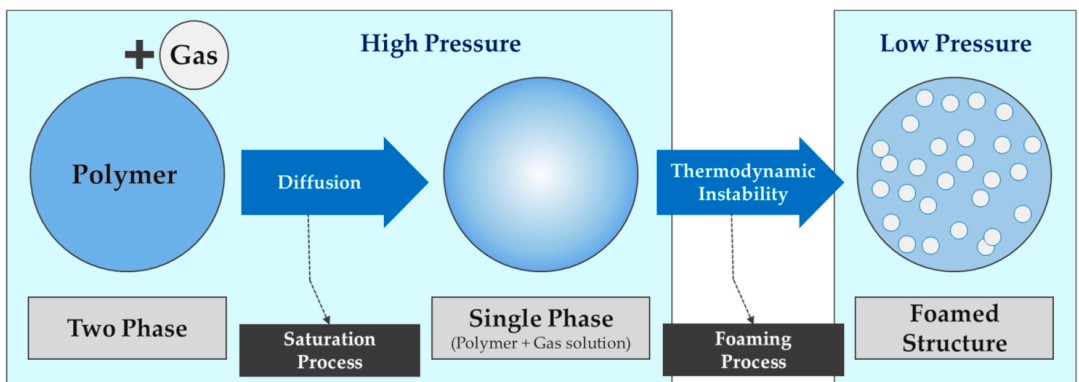

**Figure 1.** Schematic diagram of the microcellular foaming principle.

1. The polymer is saturated or impregnated with a supercritical fluid at sufficiently high pressure and defined temperature. At this time, reference is made to the glass transition temperature in the case of an amorphous polymer and the melting temperature in the case of a semi-crystalline polymer.
2. Thermodynamic instability is introduced into a supersaturated polymer–gas mixture by a rapid rise in temperature and a rapid decrease in pressure, at which time phase separation of the mixture occurs and cell nucleation occurs due to gas solubility.
3. Although the foam structure is gradually formed, as the gas diffuses from the polymer matrix to the already nucleated cells, the cells grow and stabilize after a certain period.
4. After that, the structure of the cell is stabilized through cooling.

In addition, the growth rate of nucleated cells is controlled by the gas diffusion rate, the corresponding temperature, and the viscoelastic behavior of the polymer in a supersaturated state, and the stabilization of the cell depends on the cell growth stress and melt strength.

The batch process, a foaming method developed by Martini in 1979, enables microcellular foaming in a simple manner. This process can achieve improved results at identical strength, fracture toughness, and insulation properties with the use of less material. The fabrication of microcellular foam varies drastically in terms of cell generation, depending on the saturating gas type, gas pressure, solubility, foaming temperature, and foaming time [17–19].

### 2.1. Preparations of Material

In general, the culture of microorganisms is carried out through shaking culture. However, considering the characteristics of the bacteria, the static culture method and the shaking culture method were performed simultaneously, and the resulting bacterial celluloses were compared and adopted as a material manufacturing method.

When using the static culture method, it was possible to obtain bacterial cellulose in a more solid, robust, and intact form than the shaking culture method, and it could be obtained in the form of a film rather than a spherical form.

In order to prepare a specimen for the microcellular foaming process through bacterial culture, the *Komagataeibacter xylinum* (ATCC 23767) strain was distributed from the KCTC (Korean Collection for Type Cultures, Jeollabuk-do, Republic of Korea). First, the freeze-dried bacteria ampoule was regenerated by diluting sterile PBS (Phosphate-Buffered Saline). In addition, agar medium and broth medium, two types of HS (Hestrin–Schramm) medium, were used for the ideal growth of the bacteria to be used.

The sterilized agar medium was dispensed in about 1/3 of a φ 9 size Petri dish and dried 15 to 20 min. The regenerated bacteria were streaked on the prepared agar medium three times and then cultured for 2 days in a normal incubator at 30 °C. Colonies can be used for static culture after growing to about 2 mm in size. Then, after dispensing 250 mL of broth medium in a sterilized beaker, colonies were inoculated with 5% *v/v* for static culture. Likewise, bacterial cellulose was obtained by performing static culture in an incubator at 30 °C for 12 days. All procedures were performed in the clean bench of aseptic conditions to avoid contamination with other bacteria (Table 1, Figures 2–5) [20,21].

**Table 1.** Conditions of main culture.

| Bacterial Cellulose | |
|---|---|
| Strain | *Komagataeibacter xylinum* |
| Media type | HS (Hestrin–Schramm) |
| Culture type | Static culture |
| Culture temperature [°C] | 30 ± 1 |
| Culture time [Day] | 12 |
| Cleaning media | DI water/NaOH (10%) |
| Cleaning time [Day] | 1 |

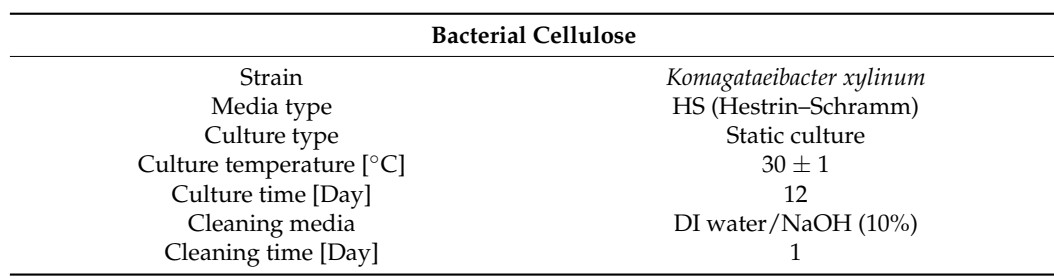

**Figure 2.** Formation of bacterial cellulose.

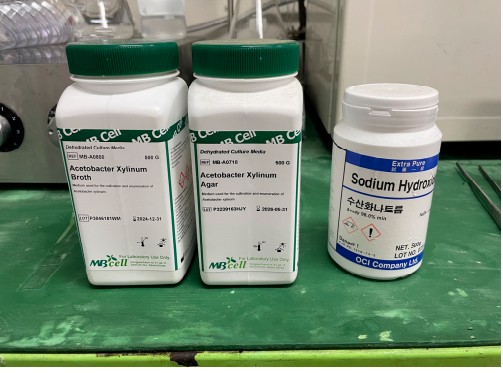

**Figure 3.** Items used for producing bacterial cellulose: agar and broth mediums, NaOH powder.

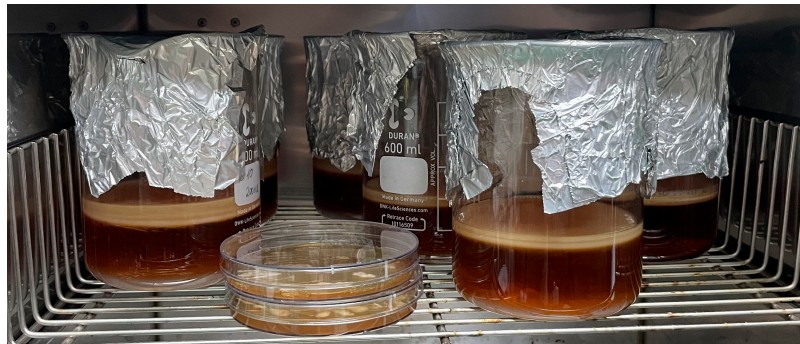

**Figure 4.** Items used for producing bacterial cellulose: Petri dishes with colonies and beakers with bacterial cellulose in an incubator.

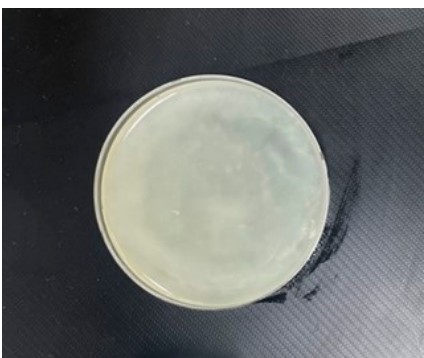

**Figure 5.** The final pellicle of bacterial cellulose.

Bacterial cellulose pellicles grown to a thickness of about 2 mm were washed with DI water and 10% NaOH solution before being used for foaming. This procedure is intended to cleanly remove any bacteria that may remain on the microcellular foaming material.

### 2.2. Experiments

A batch method was adopted to create small cells and high-density samples. The specimens were saturated with carbon dioxide (Samheung, Seoul, Republic of Korea; product grade no. $CO_2$) in a vessel with an inner diameter of 52 mm and a height of 200 mm, which was equipped with an electric heater. After the bacterial cellulose pellicle was placed in a vessel heated to 120 °C, carbon dioxide was injected at 10 MPa to create supercritical conditions.

The saturation and foaming process was performed using pressure and temperature for 4 h. A rapid pressure drop through the nozzle created a state of maximum pressure change. Finally, considering the characteristics of bacterial cellulose with a high glass transition temperature, micro-sized cell foaming was proceeded by cooling through dry ice at cryogenic temperatures (Table 2, Figure 6) [22].

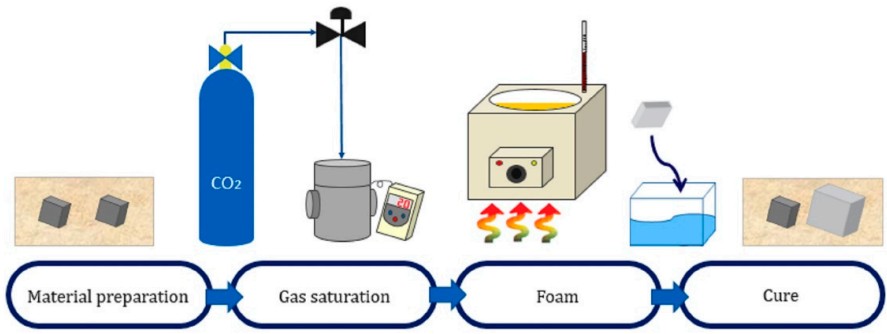

**Figure 6.** Procedure of batch process.

**Table 2.** Experimental conditions of batch process.

| Experimental Conditions | |
|---|---|
| Material | Bacterial cellulose |
| Saturation gas | $CO_2$ |
| Saturation pressure [MPa] | 10 |
| Saturation time [h] | 240 |
| Saturation/Foam temperature [°C] | 120 |
| Cooling media | Dry ice |
| Cooling temperature [°C] | −73 |
| Cooling time [s] | 120 |
| Room temperature [°C] | 20 ± 3 |

The saturation amount was measured by measuring the weight of the specimen before and after saturation. An electronic scale (OHAUS, Model No. AR2130) was used to measure the weights of the specimens. The solubility of the specimens was determined using the following equation:

$$\text{Solubility}(\%) = \frac{\text{Weight}^{\text{Gas out}} - \text{Weight}^{\text{Gas in}}}{\text{Weight}^{\text{Gas in}}} \times 100 \tag{1}$$

The foaming ratio was measured by the density of the specimen before and after the microcellular foaming process. An electronic densimeter (Alfa Mirage, Model No. MD-300S) was used to measure the densities of the specimens. The density measured using ASTM D792-20 was calculated as the foaming ratio using the following equation:

$$\text{Foaming ratio}(\%) = \frac{\text{Density}^{\text{Before}} - \text{Density}^{\text{After}}}{\text{Density}^{\text{Before}}} \times 100 \tag{2}$$

*2.3. Test and Analysis*

The expansion of the appearance can recognize the cell growth of the microcellular foam and can be accurately confirmed using scanning electron microscopy (SEM; JEOL Ltd., Peabody, MA, USA, FE-SEM Model no. IT-500). To conduct an SEM analysis, we froze the foamed specimens using liquid nitrogen (Supplied by Samheung, Seoul, Republic of Korea) and subsequently broke and pretreated them to aid clear visibility of the cross-section. The processed specimens were photographed after plasma processing (Cressington Scientific Instruments Ltd., Watford, UK, Sputter coaters Model no. Cressington 108 auto) for approximately 120 s. The confocal LSM images were taken to confirm that micro-sized cells and the surface were modified due to the microcellular foaming process.

Micro-sized cells that did not exist before foaming were created during the microcellular foaming process. Cell size was measured using Image J, and the following equation expresses the relationship between the cell size, density, and void fraction:

$$V_f = \left(\frac{\pi}{6}\right)d^3 N_f \tag{3}$$

and,

$$N_0 = \frac{N_f}{\left(1 - V_f\right)} \tag{4}$$

where $N_0$ is the cell density of the foamed specimen, $d$ is the cell size, and $V_f$ is the void fraction.

These were prepared in a rectangular shape (120 mm × 40 mm) and a thickness of 1.5 mm. In addition, a tribology test was conducted to compare the friction and wear characteristics before and after the microcellular foaming. The material was intended to be used as a support in the body, and the test conditions were applied after converting it according to the size of the specimen based on the load applied to the knee cartilage.

It was considered that the reciprocating motion is the most similar to the motion method of the human body joints among numerous methods: pin-on-disk, pin-on-reciprocating, and pin-on-cylinder. Among the three ways, the test was conducted with the pin-on-reciprocating method, which is a similar process that occurs in joint motion.

Pin-on-reciprocating tribometer was normally used to evaluate the tribological performances of the samples. The pin-on-reciprocating tribometer consists of a pin, a linear guide, a disc, a load sensor, and a load arm for load application (Table 3, Figures 7 and 8).

**Table 3.** Experimental conditions of tribology test.

| Pin-on-Reciprocating Tribometer | |
| --- | --- |
| Normal load [N] | 10 |
| Sliding speed [mm/s] | 4 |
| Sliding stroke [mm] | 8 |
| Cycles [cycle] | 300 |
| Ball size [mm] | 5 |
| Sample thickness [mm] | 1.5 |

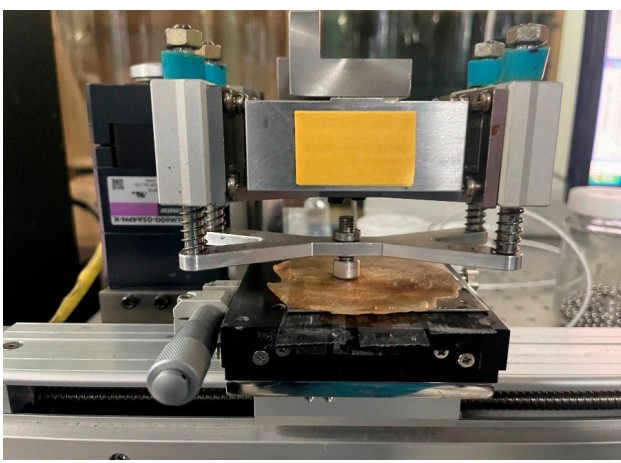

**Figure 7.** Pin-on-reciprocating-type tribology tester.

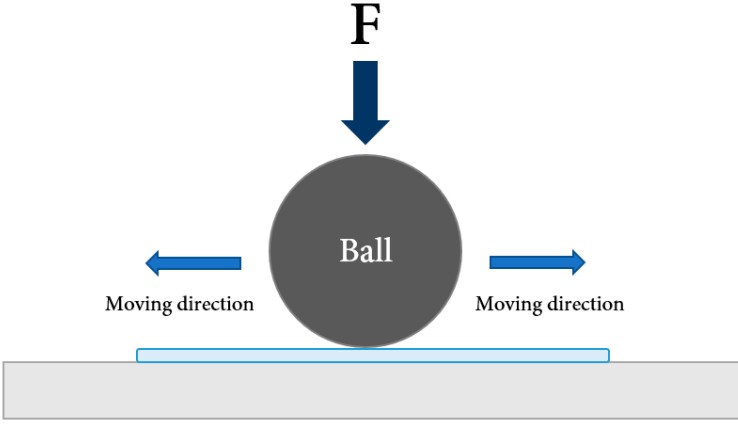

**Figure 8.** Schematic of tribology test configuration.

## 3. Results

The microcellular foaming process under subcritical conditions did not affect the foaming of bacterial cellulose with many hydroxyl groups and high glass transition temperature. However, the microcellular foaming process under supercritical conditions (10 MPa, 120 °C) makes the cells of bacterial cellulose and specimens stickier and stiffer.

In order to more closely observe the changes in the microcellular foaming cellulose with a final expansion ratio of 5.74%, SEM images and confocal LSM images were used to show internal and surface changes. Cells with an average size of 20 μm were formed on the material, and surface changes due to pores were observed (Table 4, Figures 9–11).

**Table 4.** Results of the fabrication of microcellular foamed bacterial cellulose.

| Microcellular Foamed Bacterial Cellulose | |
| --- | --- |
| Solubility [%] | 12 |
| Density [g/cm$^3$] | 0.944 |
| Foaming ratio [%] | 5.74 |
| Average of cell size [μm] | 21.1 |
| Coefficient of friction | 0.25 |

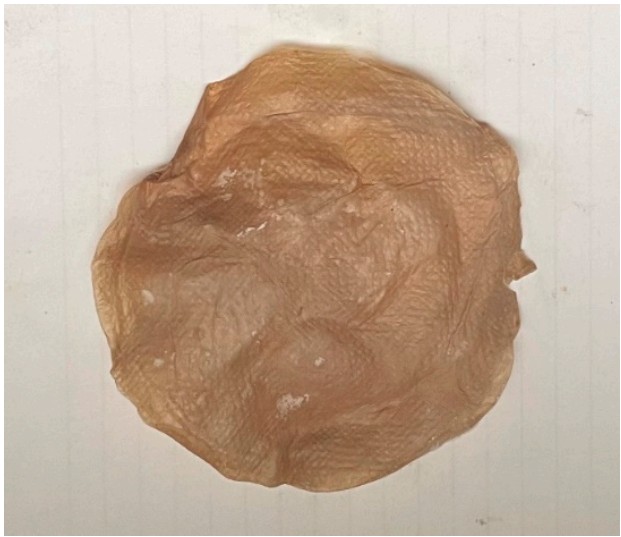

**Figure 9.** The microcellular foamed bacterial cellulose.

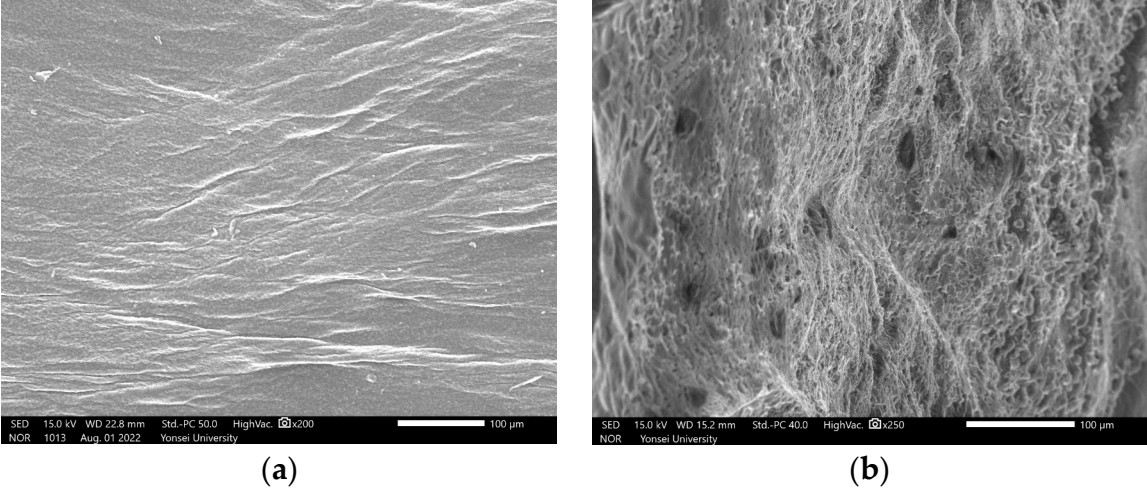

**(a)**                                        **(b)**

**Figure 10.** Scanning electron micrograph of bacterial cellulose specimens: (**a**) non-foamed bacterial cellulose (at a magnification of ×200); (**b**) microcellular foamed bacterial cellulose (at a magnification of ×250).

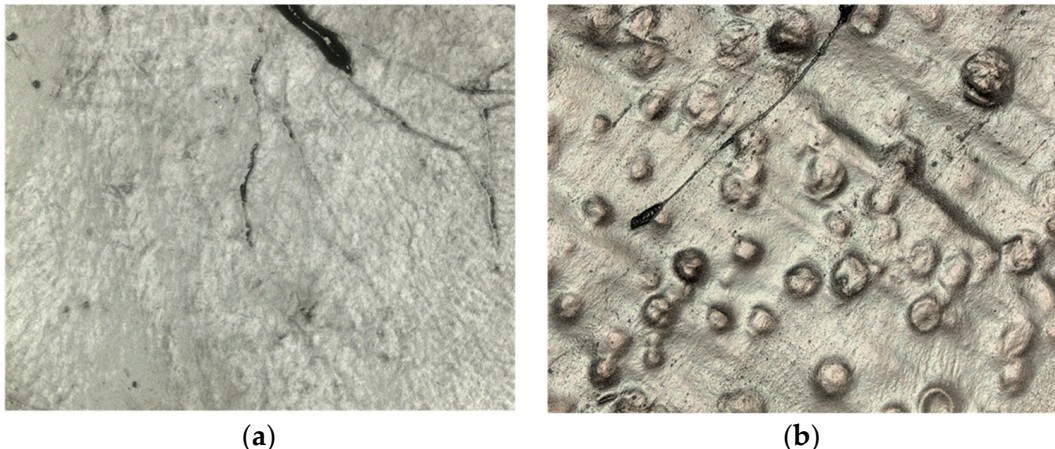

(**a**)                  (**b**)

**Figure 11.** The confocal LSM of bacterial cellulose specimens: (**a**) non-foamed bacterial cellulose; (**b**) microcellular foamed bacterial cellulose.

The coefficient of friction (COF) describes the sliding resistance of two surfaces in contact with each other and is a dimensionless number defined as the ratio between the friction force and the normal force. This value must be between 0 and 1, with higher values indicating greater resistance to slipping. Depending on its properties and surface roughness, a material with a COF less than 0.1 is considered a lubricating material.

The non-foamed bacterial cellulose specimens were broken at about 100 cycles, and the coefficient of friction also gradually increased as the cycle increased. However, microcellular foamed bacterial cellulose specimens could withstand stable frictional behavior of less than 0.3 for more than 300 cycles (Figure 12).

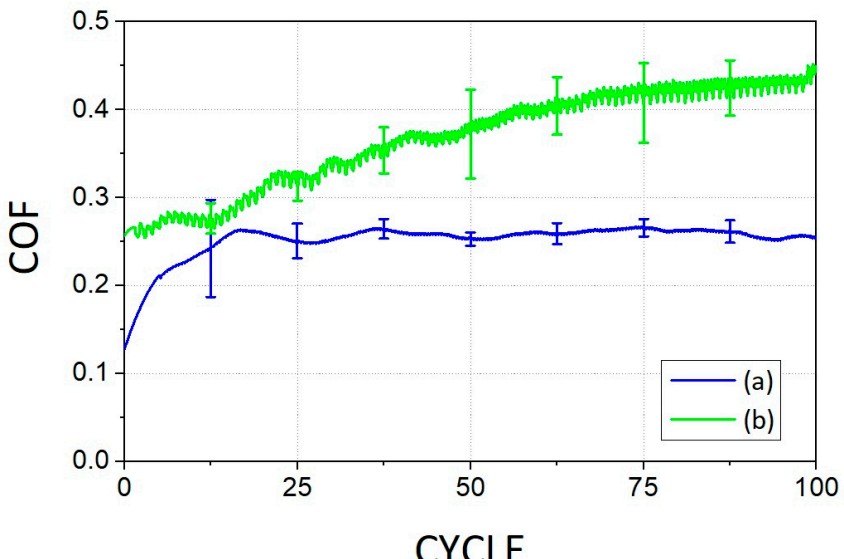

**Figure 12.** The results of tribotest: (**a**) microcellular foamed bacterial cellulose; (**b**) non-foamed bacterial cellulose.

## 4. Discussion

To improve the friction or wear properties of most materials, researchers perform surface modification or use lubricants. Typical examples include coating, surface roughness control, and surface pattern formation. In addition, there are methods such as heat treatment, anodizing, and shot peening, which increase the hardness of the surface and reduce wear [23–27].

On the other hand, bacterial cellulose is an organic polymer with abundant hydroxyl groups. In the case of bacterial cellulose covered in this study, it is difficult to modify the surface using the method mentioned earlier due to the large number of hydroxyl groups in materials. Therefore, surface modification was attempted using a new technology, the microcellular foaming process.

Using the batch process, one of the microcellular foaming processes, supercritical fluid ($CO_2$) is saturated. Foaming is carried out at the same time. The temperature change is by the cooling method through dry ice using the characteristics of materials with a high glass transition temperature (Tg). Implementing a cell and modifying the physical properties even in bacterial cellulose was possible.

Microcellular foaming under subcritical conditions did not affect the foaming of bacterial cellulose containing many hydroxyl groups and having a high glass transition temperature. This proves the microcellular foaming theory that gas saturation is possible using the supercritical fluid state and that a large pressure difference and temperature difference affect the phase change in the polymer and promote foaming.

Bacterial cellulose produced from the metabolism of acetic acid bacteria foams the material without using chemical substances, such as foaming agents, so that biocompatibility, biodegradability, and blood compatibility are not affected, whereas stabilizing the coefficient of friction and pores with an average size of 200 μm are securable. In addition, the time required for fabrication could be reduced by simultaneously performing gas saturation and foaming processes in a supercritical fluid state.

As it was confirmed that there is a sufficient difference in the foaming ratio or cell size by controlling the pressure and temperature without chemical additives, the possibility of having other characteristics by adjusting the parameters in the pretreatment process or changing the post-treatment method is suggested.

Until now, studies using bacterial cellulose as a base material have been insignificant due to difficulties in modifying mechanical properties and difficulties in mass production. Therefore, it is considered that the top priority is to identify suitable technologies to be applied to material property modification. Material development using the microcellular foaming process reduces product production time and is representative of an environmentally friendly method. Therefore, if new characteristics are identified by applying various foaming methods, the scope of application will gradually expand.

This research will further develop cell formation and cell density control technology inside advanced block copolymers, optimize the physical properties of materials, activate the potential of cellular structures, and open new approaches. In addition, it is necessary to develop it for potential use by considering new applications and applications essential for future industrialization, such as the trade-off between transparency and insulation, improvement of the glass transition temperature, and development of continuous processes for micro/nanocellular foams.

In addition to biodegradability, the biomaterial scaffold must be able to equally describe the composition and structure of the articular cartilage extracellular matrix (ECM) to support cell adhesion and smooth movement. Therefore, biocompatibility (ISO 10993) tests are currently underway for the material, and characteristics related to cell toxicity, cell attachment, migration, and proliferation can be identified. Furthermore, finally, by incorporating cellulose bonds into areas that cannot be repaired, we will present a plan to utilize them as scaffolds and human dialysis filters by identifying the function as a target treatment scaffold and identifying newly emerging characteristics.

## 5. Conclusions

Microcellular foams under supercritical conditions successfully enabled the foaming of bacterial cellulose with high glass transition temperature and high hydroxyl groups. Saturation and foaming of 10 MPa carbon dioxide at 120 °C for 4 h were carried out simultaneously in a high-pressure vessel, and microcellular foamed bacterial cellulose specimens were produced by cooling using dry ice at the same time as depressurization.

Microcellular foamed bacterial cellulose showed differences in color change, surface waviness, increase in tensile force, change in thickness, and decrease in coefficient of friction. In the microcellular foamed bacterial cellulose with a final foaming ratio of 5.74%, cells with an average of 20 μm were formed in the material. The friction coefficient after foaming was 0.25 compared to before foaming. It was confirmed that the wear cycle can also withstand more than 300 cycles in 50 cycles.

The result means that surface modification is possible by foaming bacterial cellulose using the microcellular foaming process, and the stable COF (coefficient of friction) numbers indicate that stable behavior can be achieved when used as an artificial joint replacement material in the human body. Also, it suggests the need for more in-depth research on the utilization of bacterial cellulose.

**Author Contributions:** Conceptualization, J.H.; methodology, J.H., D.K. and G.Y.; software, J.H. and J.-H.H.; validation, J.H. and J.-H.H.; formal analysis, J.H.; investigation, J.H. and J.-H.H.; resources, J.H. and G.Y.; data curation, J.H.; writing—original draft preparation, J.H.; writing—review and editing, J.H.; visualization, J.H. and J.-H.H.; supervision, S.W.C.; project administration, J.H. and K.H.K. All authors have read and agreed to the published version of the manuscript.

**Funding:** This research received no external funding.

**Data Availability Statement:** The data presented in this study are available upon request from the first author.

**Acknowledgments:** The first author sincerely thanks Sung Woon Cha and other authors.

**Conflicts of Interest:** The authors declare no conflict of interest.

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
