# Peer review of "Friction and Wear Characteristics of Bacterial Cellulose Modified by Microcellular Foaming Process"

_lubricants, doi:10.3390/lubricants11080322_

Round 1

Reviewer 1 Report

The authors studied the friction and wear characteristics of bacterial cellulose modified by microcellular foaming process. There are problems should be noticed as following:

(1)  Komagataeibacter Xylinus is a company or other method? It should be explained or deleted in abstract.

(2) The authors mainly studied the friction and wear characteristics of bacterial cellulose. But there is few description about the friction and wear characteristics of bacterial cellulose. Is there no related study about it? It suggests to review the related references in section of introduction.   

(3) There are some style errors in the text.

(4) It do not need to describe too more background knowledge in the section of Materials and Methods. It should describe the method and materials.

(5)  Why the authors used the Pin -on-reciprocating type tribology tester to analyze the the friction and wear characteristics.

(6) COF should be explained in Figure 10, and the different lines should be explained.  

(7) The process parameters should be studied in the manuscript. It is too simple for the author to only study the COF of prepared bacterial cellulose. The comparison of different parameters ans the mechanism of anti-friction are better included in the study. The prepared the Bacterial Cellulose Modified by Microcellular Foaming Process is beneficial or harmful to the friction and wear? And why?

Author Response

Review Report

Dear: Reviewer (1)

Thank you very much for reviewing our paper. We have written an answer to the reviewer's opinion and revised the manuscript by reflecting on this. We have reflected on all the reviews of the reviewers and have improved the quality of all manuscript figures. In addition, we will receive an English proofing service if the paper is accepted after the revision process has ended. We have marked the revised manuscript and added red colored memo.

Q1. Komagataeibacter Xylinus is a company or other method? It should be explained or deleted in abstract.

(Author): Komagataeibacter Xylinus is a species of bacteria best known for its ability to produce cellulose, specifically bacterial cellulose. The species was first described in 1886 by Adrian John Brown, who identified the bacteria while studying fermentation. After further studies, the final scientific name was confirmed as Komagataeibacter Xylinus (Yamada et al. 2013).

We obtain materials necessary for experiments through this strain. Worldwide, Komagataeibacter Xylinus has been given ATCC 23767 number in the American Type Culture Collection.

Scientific classification

Domain

Bacteria

Phylum

Pseudomonadota

Class

Alphaproteobacteria

Order

Rhodospirillales

Family

Acetobacteraceae

Genus

Komagataeibacter

Species

K. xylinus

Binomial name

Komagataeibacter xylinus (Yamada et al. 2013)

[Revised text, Line 87-95]

Acetobacter Xylinum (or Gluconacetobacter Xylinus) was first described in 1886 by Adrian John Brown, who identified the bacterium while studying fermentation. Later, in 2013 by Yamada et al., a new genus Komagataeibacter was established and named Komagataeibacter Xylinum (American Type Culture Collection, ATCC 23767). This is a species of bacteria best known for its ability to produce cellulose, specifically bacterial cellulose. Also, the strain is a member of acetic bacteria, a group of gram-negative aerobic bacteria that produce acetic acid during fermentation. Bacterial cellulose is involved in biofilm formation. It is chemically identical to plant cellulose but has a different physical structure and properties.

Q2. The authors mainly studied the friction and wear characteristics of bacterial cellulose. But there is few description about the friction and wear characteristics of bacterial cellulose. Is there no related study about it? It suggests to review the related references in section of introduction.  

(Author): Bacterial cellulose is a material usually dealt with in the field of food and nutrition and has recently been introduced as an item that can be added to lubricants [1-2].

Until now, bacterial cellulose is only widely known as a material with infinite potential that can be used as a material for the human body. In other words, many studies are being conducted, but no results have been derived. Therefore, nothing is said about the frictional and abrasion properties of bacterial cellulose without additives, mostly in terms of tensile and compressive properties. (It was first covered in that article.) However, for the same reason as your opinion, there may be recent new journals or research results, so an additional search was conducted. And it was difficult to find a study that fit or same with this object.

[1] Gorbacheva, S.; Yadykova, A.; Ilyin, S. Rheological and tribological properties of low-temperature greases based on cellulose acetate butyrate gel. Carbohydrate Polymers, 2021, 272, 118509.

[2] Fuadi, Z.; Rahmadiawan, D.; Kurniawan, R.; Mulana, F.; Abral, H.; Nasruddin, N.; Khalid, M. Effect of Graphene Nanoplatelets on Tribological Properties of Bacterial Cellulose/Polyolester Oil Bio-Lubricant. Frontiers in Mechanical Engineering, 2022, 8-2022, 810847.

Q3. There are some style errors in the text.

(Author): Thank you very much for your helpful comments. The overall content, such as alignment of image files in the document, spacing, and paragraphs, has been modified. In addition, grammatical advice was obtained through a professional proofreading company. In addition, we will complete the professional article through negotiations with the editor before the final journal publication.

Q4. It do not need to describe too more background knowledge in the section of Materials and Methods. It should describe the method and materials.

(Author): Concerning the comment, the history related to bacterial cellulose was moved to the introduction section. In Preparations of Material, more detailed methods and materials were described.

[Deleted text, Line 135-141]

[Revised text, Line 142-166]

In general, the culture of microorganisms is carried out through shaking culture. However, considering the characteristics of the bacteria, the static culture method and the shaking culture method were performed simultaneously, and the resulting bacterial celluloses were compared and adopted as a material manufacturing method.

When using the static culture method, it was possible to obtain bacterial cellulose in a more solid, robust, and intact form than the shaking culture method, and it could be obtained in the form of a film rather than a spherical form.

In order to prepare a specimen for the microcellular foaming process through bacterial culture, the Komagataeibacter Xylinum (ATCC 23767) strain were distributed from the KCTC (Korean Collection for Type Cultures). First, the freeze-dried bacteria ampoule is re-generated by diluting sterile PBS (Phosphate-Buffered Saline). In addition, agar medium and broth medium, two types of HS (Hestrin-Schramm) medium, were used for the ideal growth of the bacteria to be used.

The sterilized agar medium was dispensed in about 1/3 of a φ9 size petri dishes and dried 15 to 20 minutes. The regenerated bacteria were streaked on the prepared agar medium three times and then cultured for 2 days in a normal incubator at 30 °C. Colonies can be used for static culture after growing to about 2 mm in size. Then, after dispensing 250 ml of broth medium in a sterilized beaker, colonies were inoculated with 5 % v/v for static culture. Likewise, bacterial cellulose was obtained by performing static culture in an incubator at 30 °C for 12 days. All procedures were performed in the clean bench of aseptic conditions to avoid contamination with other bacteria and the sterilization was done by the autoclave (Table 1., Figure 2., Figure 3., Figure 4., Figure 5.) [20-21].

Q5. Why the authors used the Pin -on-reciprocating type tribology tester to analyze the friction and wear characteristics.

(Author): The friction test determines the type of experiment according to the movement method and scale. We fabricated the specimen considering the force applied to the actual human knee cartilage and the size of the knee cartilage. (Standard - based on adult female 65kg, maximum load: when going uphill, 310N) Considering the movement direction and contact shape were the most similar to the motion method of the human body joints among numerous methods, we finally considered the methods of pin-on-disk, pin-on-reciprocating and pin-on-cylinder.

Among the three ways, the test was conducted with the pin-on-reciprocating method, which is more similar to the process that occurs in joint motion.

[Revised text, Line 231-238]

It was considered that the reciprocating motion is the most similar to the motion method of the human body joints among numerous methods, pin-on-disk, pin-on-reciprocating, and pin-on-cylinder. Among the three ways, the test was conducted with the pin-on-reciprocating method, which is more similar process that occurs in joint motion.

Pin-on-reciprocating tribometer was normally used to evaluate the tribological performances of the samples. The pin-on-reciprocating tribometer consists of a pin, a linear guide, a disc, a load sensor, and a load arm for load application (Table 3., Figure 7., Figure 8.).

Q6. COF should be explained in Figure 10, and the different lines should be explained. 

(Author): The coefficient of friction (COF) describes the sliding resistance of two surfaces in contact with each other and is a dimensionless number defined as the ratio between the friction force and the normal force. This value must be between 0 and 1, with higher values indicating greater resistance to slipping. Depending on its properties and surface roughness, a material with a COF less than 0.1 is considered a lubricating material. By mentioning the contents in the text, we improved readers' understanding of COF.

In the experimental results, non-foamed bacterial cellulose fractured in about 100 cycles, and foamed bacterial cellulose did not break even after 300 cycles or more. Therefore, we have graphed up to 100 cycles for practical comparison. In addition, visibility was secured by calculating the final result graphs (Figure 10.) as average values and plotting them in one chart.

[Revised text, Line 270-274]

The coefficient of friction (COF) describes the sliding resistance of two surfaces in contact with each other and is a dimensionless number defined as the ratio between the friction force and the normal force. This value must be between 0 and 1, with higher values indicating greater resistance to slipping. Depending on its properties and surface roughness, a material with a COF less than 0.1 is considered a lubricating material.

[Deleted text, Line 279-282]

[Revised text, Line 283-285]

Figure 12. The results of tribotest: (a) Microcellular foamed bacterial cellulose; (b) Non-foamed bacterial cellulose.

Q7. The process parameters should be studied in the manuscript. It is too simple for the author to only study the COF of prepared bacterial cellulose. The comparison of different parameters and the mechanism of anti-friction are better included in the study. The prepared the Bacterial Cellulose Modified by Microcellular Foaming Process is beneficial or harmful to the friction and wear? And why?

(Author): Various methods are applied to improve the friction and wear characteristics of the material to be used. Methods include coating, surface roughness control, surface pattern formation, heat treatment, anodizing, and shot peening. Therefore, it is a good research direction to try various methods and compare the result values according to your opinion. However, modifying the surface of bacterial cellulose, an organic polymer with many hydroxyl groups, was impossible by the method mentioned above.

Bacterial cellulose, which appears as a material in the study, is mostly processed by heat drying or freeze-drying to make powder and then used as a lubricant. In this study, we are trying to see the characteristics of intact cellulose without processing, and we are continuously checking various properties such as tension, compression, and biocompatibility.

[Revised text, line 287-296]

To improve the friction or wear properties of most materials, researchers perform surface modification or use lubricants. Typical examples include coating, surface rough-ness control, and surface pattern formation. In addition, there are methods such as heat treatment, anodizing, and shot peening, which increase the hardness of the surface and reduce wear.

On the other hand, bacterial cellulose is an organic polymer with abundant hydroxyl groups. In the case of bacterial cellulose covered in this study, it is difficult to modify the surface using the method as mentioned earlier due to the large number of hydroxyl groups in materials. Therefore, surface modification was attempted using a new technology, the microcellular foaming process.

Reviewer 2 Report

The paper presents a series of information about the possibility of producing a sheet of bacterial cellulose that can be used as a support applicable to the human body

The paper may be of interest to the scientific community through the topic addressed.

Authors should consider the following observations:

- The introduction needs to be substantially improved as it presents very little information on how a bacterial cellulose sheet can be used to reduce joint friction. Also, at the end of the introduction section, the structure of the article and the main objective of the research should be presented in more detail.

- A detailed description of the technology for obtaining bacterial cellulose foils is required. It is also necessary to present a macroscopic image of the installation used to obtain the foils.

- The research methodology must be detailed. In this sense, a program of experimental research must be presented that takes into account several levels for the adjusted parameters (pressing force, rotation speed).

- The results presented graphically in the Figure need to be much detailed and developed. In the form presented, the article is more of a technical report without identifying the scientific part. I don't understand what the point is that such a film can withstand 300 cycles.

- The obtained results must be discussed in detail to highlight their novelty in relation to other research in the field.

- The conclusions are too general without specifying the practical applications of the research. Also, at the end of the conclusions, future research directions should be specified.

Author Response

Review Report

Dear: Reviewer (2)

Thank you very much for reviewing our paper. We have written an answer to the reviewer's opinion and revised the manuscript by reflecting on this. We have reflected on all the reviews of the reviewers and have improved the quality of all manuscript figures. In addition, we will receive an English proofing service if the paper is accepted after the revision process has ended. We have marked the revised manuscript and added red colored memo.

Q1. The introduction needs to be substantially improved as it presents very little information on how a bacterial cellulose sheet can be used to reduce joint friction. Also, at the end of the introduction section, the structure of the article and the main objective of the research should be presented in more detail.

(Author): Due to the current increase in the elderly population, information on the development and improvement of biomedical polymers required by the medical field has been added. In addition, we tried to increase readers' understanding by explaining why bacterial cellulose was adopted as a foam material and its advantages and benefits.

[Revised text, Line 42-52]

Recently, due to the rapid increase in the elderly population, degenerative diseases are making many patients sick worldwide, and these diseases have reduced the quality of human life. In particular, the number of patients undergoing surgery due to degenerative arthritis is increasing by 10% yearly. The knee area is the most representative of artificial joint replacement among them. Many seniors are looking for a new life by inserting an artificial joint made of materials such as ceramic and metal into the body.

However, there was pain due to the shape mismatch between the knee and the artificial joint and discomfort due to the stiffness of the inserted joint. In addition, a bigger problem is the lifespan of artificial joints, which are limited to 10 years, and the problem of necrosis of surrounding tissues due to debris generated in the body due to long-term use of the artificial joint.

[Revised text, Line 77-86]

Bacterial cellulose has excellent physical properties such as high absorption, elasticity, moisture retention, crystallinity, and oxygen transfer efficiency, which were difficult to find in vegetable cellulose [14-15]. However, it is difficult to commercialize due to high production costs and difficulty in mass production. However, its applicability to biomedical and electronic materials is highly evaluated due to its high biocompatibility, flexibility, moisture content, and environmental friendliness. Moreover, currently, it is expanding into various product groups such as medical mats using high binding capacity, speaker diaphragms using high elastic modulus, thickening agents, swelling agents, and mask packs using high absorbency. It is being used as a high-value-added product as a biopolymer.

Q2. A detailed description of the technology for obtaining bacterial cellulose foils is required. It is also necessary to present a macroscopic image of the installation used to obtain the foils.

(Author): The production process of bacterial cellulose was written in more detail, and pictures of actual experiment supplies were added separately from the process schematic.

[Revised text, Line 142-169]

In general, the culture of microorganisms is carried out through shaking culture. However, considering the characteristics of the bacteria, the static culture method and the shaking culture method were performed simultaneously, and the resulting bacterial celluloses were compared and adopted as a material manufacturing method.

When using the static culture method, it was possible to obtain bacterial cellulose in a more solid, robust, and intact form than the shaking culture method, and it could be obtained in the form of a film rather than a spherical form.

In order to prepare a specimen for the microcellular foaming process through bacterial culture, the Komagataeibacter Xylinum (ATCC 23767) strain were distributed from the KCTC (Korean Collection for Type Cultures). First, the freeze-dried bacteria ampoule is regenerated by diluting sterile PBS (Phosphate-Buffered Saline). In addition, agar medium and broth medium, two types of HS (Hestrin-Schramm) medium, were used for the ideal growth of the bacteria to be used.

The sterilized agar medium was dispensed in about 1/3 of a φ9 size petri dishes and dried 15 to 20 minutes. The regenerated bacteria were streaked on the prepared agar medium three times and then cultured for 2 days in a normal incubator at 30 °C. Colonies can be used for static culture after growing to about 2 mm in size. Then, after dispensing 250 ml of broth medium in a sterilized beaker, colonies were inoculated with 5 % v/v for static culture. Likewise, bacterial cellulose was obtained by performing static culture in an incubator at 30 °C for 12 days. All procedures were performed in the clean bench of aseptic conditions to avoid contamination with other bacteria (Table 1., Figure 2., Figure 3., Figure 4., Figure 5.) [20-21].

Bacterial cellulose pellicles grown to a thickness of about 2 mm were washed with DI water and 10% NaOH solution before being used for foaming. This procedure is intended to cleanly remove any bacteria that may remain on the microcellular foaming material.

[Revised text, Line 174-180]

Figure 3. Items used for producing bacterial cellulose – Agar and broth medium, NaOH powder.

Figure 4. Items used for producing bacterial cellulose – Petri dishes with colonies, and beakers with bacterial cellulose in an incubator.

Q3. The research methodology must be detailed. In this sense, a program of experimental research must be presented that takes into account several levels for the adjusted parameters (pressing force, rotation speed).

(Author): The friction test determines the type of experiment according to the movement method and scale. We fabricated the specimen considering the force applied to the actual human knee cartilage and the size of the knee cartilage. (Standard - based on adult female 65kg, maximum load: when going uphill, 310N) Considering that the reciprocating motion is the most similar to the motion method of the human body joints among numerous methods, we finally considered the methods of pin-on-disk, pin-on-reciprocating, and pin-on-cylinder.

Among the three ways, the test was conducted with the pin-on-reciprocating method, which is more similar to the process that occurs in joint motion.

[Revised text, Line 231-238]

It was considered that the reciprocating motion is the most similar to the motion method of the human body joints among numerous methods, pin-on-disk, pin-on-reciprocating, and pin-on-cylinder. Among the three ways, the test was conducted with the pin-on-reciprocating method, which is more similar process that occurs in joint motion.

Pin-on-reciprocating tribometer was normally used to evaluate the tribological performances of the samples. The pin-on-reciprocating tribometer consists of a pin, a linear guide, a disc, a load sensor, and a load arm for load application (Table 3., Figure 7., Figure 8.).

Q4. The results presented graphically in the Figure need to be much detailed and developed. In the form presented, the article is more of a technical report without identifying the scientific part. I don't understand what the point is that such a film can withstand 300 cycles.

(Author): The coefficient of friction (COF) describes the sliding resistance of two surfaces in contact with each other and is a dimensionless number defined as the ratio between the friction force and the normal force. This value must be between 0 and 1, with higher values indicating greater resistance to slipping. Depending on its properties and surface roughness, a material with a COF less than 0.1 is considered a lubricating material. By mentioning the contents in the text, we improved readers' understanding of COF.

In the experimental results, non-foamed bacterial cellulose fractured in about 100 cycles, and foamed bacterial cellulose did not break even after 300 cycles or more. Therefore, we have graphed up to 100 cycles for practical comparison. In addition, visibility was secured by calculating the final result graphs (Figure 12.) as average values and plotting them in one chart.

This result means that surface modification is possible by foaming bacterial cellulose using the microcellular foaming process, and the stable COF (coefficient of friction) numbers indicate that stable behavior can be achieved when used as an artificial joint replacement material in the human body.

[Revised text, Line 270-278]

The coefficient of friction (COF) describes the sliding resistance of two surfaces in contact with each other and is a dimensionless number defined as the ratio between the friction force and the normal force. This value must be between 0 and 1, with higher values indicating greater resistance to slipping. Depending on its properties and surface roughness, a material with a COF less than 0.1 is considered a lubricating material.

The non-foamed bacterial cellulose specimens were broken at about 100 cycles, and the coefficient of friction also gradually increased as the cycle increased. However, microcellular foamed bacterial cellulose specimens could withstand stable frictional behavior of less than 0.3 for more than 300 cycles (Figure 12.).

[Deleted text, Line 279-282]

[Revised text, Line 283-285]

Figure 12. The results of tribotest: (a) Microcellular foamed bacterial cellulose; (b) Non-foamed bacterial cellulose.

Q5. & Q6. The obtained results must be discussed in detail to highlight their novelty in relation to other research in the field., The conclusions are too general without specifying the practical applications of the research. Also, at the end of the conclusions, future research directions should be specified.

(Author): Various methods are applied to improve the friction and wear characteristics of the material to be used. Methods include coating, surface roughness control, surface pattern formation, heat treatment, anodizing, and shot peening. Therefore, it is a good research direction to try various methods and compare the result values according to your opinion. However, modifying the surface of bacterial cellulose, an organic polymer with many hydroxyl groups, was impossible by the method mentioned above.

Bacterial cellulose, which appears as a material in the study, is mostly processed by heat drying or freeze-drying to make powder and then used as a lubricant. In this study, we are trying to see the characteristics of intact cellulose without processing, and we are continuously checking various properties such as tension, compression, and biocompatibility.

[Revised text, Line 287-296]

To improve the friction or wear properties of most materials, researchers perform surface modification or use lubricants. Typical examples include coating, surface roughness control, and surface pattern formation. In addition, there are methods such as heat treatment, anodizing, and shot peening, which increase the hardness of the surface and reduce wear.

On the other hand, bacterial cellulose is an organic polymer with abundant hydroxyl groups. In the case of bacterial cellulose covered in this study, it is difficult to modify the surface using the method as mentioned earlier due to the large number of hydroxyl groups in materials. Therefore, surface modification was attempted using a new technology, the microcellular foaming process.

[Revised text, Line 324-338]

Using the batch process, one of the microcellular foaming processes, supercritical fluid (COâ‚‚), is saturated. Foaming is carried out at the same time. The temperature change is by the cooling method through dry ice using the characteristics of materials with a high glass transition temperature (Tg). Implementing a cell and modifying the physical properties even in bacterial cellulose was possible.

This research will further develop cell formation and cell density control technology inside advanced block copolymers, optimize the physical properties of materials, activate the potential of cellular structures, and open new approaches. In addition, it is necessary to develop it for potential use by considering new applications and applications essential for future industrialization, such as the trade-off between transparency and insulation, improvement of the glass transition temperature, and development of continuous processes for micro/nanocellular foams.

In addition to biodegradability, the biomaterial scaffold must be able to equally describe the composition and structure of the articular cartilage extracellular matrix (ECM) to support cell adhesion and smooth movement. Therefore, biocompatibility (ISO 10993) tests are currently underway for the material, and characteristics related to cell toxicity, cell attachment, migration, and proliferation can be identified. Furthermore, finally, by incorporating cellulose bonds into areas that cannot be repaired, we will present a plan to utilize them as scaffolds and human dialysis filters by identifying the function as a target treatment scaffold and identifying newly emerging characteristics.

[Revised text, Line 351-355]

This result means that surface modification is possible by foaming bacterial cellulose using the microcellular foaming process, and the stable COF (coefficient of friction) numbers indicate that stable behavior can be achieved when used as an artificial joint replacement material in the human body. Also, it suggests the need for more in-depth research on the utilization of bacterial cellulose.

Round 2

Reviewer 1 Report

All the comments have been revised and corrected, and now it is appropriate for publication.

Reviewer 2 Report

The authors responded to the comments made and improved the paper accordingly.